# Orbital character of the spin-reorientation transition in TbMn$_6$Sn$_6$

S. X. M. Riberolles [1], Tyler J. Slade[1,2], R. L. Dally [3], P. M. Sarte[4], Bing Li [2], Tianxiong Han [2], H. Lane[5,6], C. Stock[5], H. Bhandari[7,8], N. J. Ghimire [7,8], D. L. Abernathy [9], P. C. Canfield [1,2], J. W. Lynn [3], B. G. Ueland [1] & R. J. McQueeney[1,2] ✉

Ferromagnetic (FM) order in a two-dimensional kagome layer is predicted to generate a topological Chern insulator without an applied magnetic field. The Chern gap is largest when spin moments point perpendicular to the kagome layer, enabling the capability to switch topological transport properties, such as the quantum anomalous Hall effect, by controlling the spin orientation. In TbMn$_6$Sn$_6$, the uniaxial magnetic anisotropy of the Tb$^{3+}$ ion is effective at generating the Chern state within the FM Mn kagome layers while a spin-reorientation (SR) transition to easy-plane order above $T_{SR} = 310$ K provides a mechanism for switching. Here, we use inelastic neutron scattering to provide key insights into the fundamental nature of the SR transition. The observation of two Tb excitations, which are split by the magnetic anisotropy energy, indicates an effective two-state orbital character for the Tb ion, with a uniaxial ground state and an isotropic excited state. The simultaneous observation of both modes below $T_{SR}$ confirms that orbital fluctuations are slow on magnetic and electronic time scales < ps and act as a spatially-random orbital alloy. A thermally-driven critical concentration of isotropic Tb ions triggers the SR transition.

Two-dimensional (2D) kagome metals possess both flat electronic bands and Dirac band crossings that provide an adaptable framework for the emergence of a variety of correlated and topological phases[1]. For example, the Haldane model predicts that a Chern insulator may be generated (without an applied magnetic field) in a ferromagnetic (FM) kagome layer when spin-polarized Dirac crossings are gapped by spin-orbit coupling (SOC)[2,3]. Generating this topological phase in real (3D) materials requires that the Dirac bands are composed of orbitals with 2D character (such as $d_{x^2-y^2}$) in order to minimize coupling between kagome layers. For a 2D Dirac crossing, the Chern gap is largest when spins point perpendicular to the kagome layer.

This scenario has generated interest in $R$Mn$_6$Sn$_6$ ($R$166) compounds where the magnetic anisotropy of the rare-earth ion ($R$) plays the crucial role of controlling the spin orientation of the FM Mn kagome layers. The family of $R$166 is known for their combination of unique magnetic instabilities[4–12] and the topological character of their electronic bands[13–15]. Thus, they provide a promising framework for highly tunable magnetic topological insulators and semimetals.

[1]Ames National Laboratory, Ames, Iowa 50011, USA. [2]Department of Physics and Astronomy, Iowa State University, Ames, IA 50011, USA. [3]NIST Center for Neutron Research, National Institute of Standards and Technology, Gaithersburg, MD 20899, USA. [4]Materials Department and California Nanosystems Institute, University of California Santa Barbara, Santa Barbara, CA 93106, USA. [5]School of Physics and Astronomy, University of Edinburgh, Edinburgh EH9 3JZ, United Kingdom. [6]School of Physics, Georgia Institute of Technology, Atlanta, GA 30332, USA. [7]Department of Physics and Astronomy, George Mason University, Fairfax, VA 22030, USA. [8]Quantum Science and Engineering Center, George Mason University, Fairfax, VA 22030, USA. [9]Neutron Scattering Division, Oak Ridge National Laboratory, Oak Ridge, TN 37831, USA. ✉e-mail: mcqueeney@ameslab.gov

Hexagonal R166 compounds consist of alternating Mn kagome and R triangular layers, and starkly different properties are found for magnetic and non-magnetic R ions. For the latter, the magnetism in compounds such as Y166 is defined by FM Mn kagome layers with weak easy-plane anisotropy. Complex incommensurate helical magnetism is formed by competing FM and antiferromagnetic (AF) coupling between kagome layers, resulting in a rich field and temperature phase diagram demonstrating the topological Hall effect[6,10–12]. For magnetic R ions, strong AF R-Mn coupling provides a collinear 3D ferrimagnetic network where the orbital ground state of the R ion determines either a uniaxial (Tb), easy-cone (Ho,Dy), or easy-plane (Gd, Er) magnetic anisotropy[16]. The variability of the magnetic anisotropy for the different rare-earth ions originates from orbital ground states enabled by large fourth-order terms in the crystalline electric field (CEF) potential in the R166 structure[16]. It was recognized that the unique uniaxial magnetism of Tb166 is supremely effective at gapping spin-polarized Dirac band crossings, potentially creating a two-dimensional Chern insulator that is a realization of the spinless Haldane model[13]. However, Tb166 remains metallic, and achieving quantized Hall conductivity of a Chern insulator requires methods to control the Fermi energy by chemical substitution or gating of thin film samples.

These materials are also characterized by instabilities in the moment direction. In Tb166, a spin-reorientation (SR) transition from uniaxial to easy-plane collinear ferrimagnetism occurs above $T_{SR}$ = 310 K[17,18]. This first-order SR transition may also be driven by a magnetic field applied along the magnetic hard axis. The prospects for rare-earth engineering of R166 compounds, which utilize mixed orbital moments, anisotropies, and exchange couplings, promises to create magnetic instabilities necessary to enable switching of topological magnets[15].

To better understand the role of R orbital states and their temperature evolution, we investigate the intrinsic nature of the SR transition in Tb166 using inelastic neutron scattering (INS). This transition has been described in terms of the competition between easy-plane Mn magnetic anisotropy and a temperature-dependent uniaxial Tb anisotropy[19–22]. In this description, fluctuations of the Tb moment direction with increasing temperature weaken the uniaxial Tb anisotropy, ultimately causing the SR transition[23].

Here, our INS data provide an insightful microscopic picture of the SR transition that is consistent with this scenario. At low temperatures, the observation of a propagating Tb spin wave mode near 25 meV confirms both the strong uniaxial ground state anisotropy of the Tb ion and AF coupling between the Tb and Mn sublattices. Above $T_{SR}$, the Tb mode appears below 10 meV, which is consistent with an isotropic Tb ion. At intermediate temperatures, the simultaneous appearance of both the upper (uniaxial) and lower (isotropic) Tb modes are fingerprints of unusual Tb orbital dynamics. Within a random-phase approximation (RPA) description of the local Tb CEF states coupled to Mn, one expects a spectrum of orbital exciton modes (propagating, magnon-like quasiparticles comprised of Tb CEF excitations) to appear at high temperatures. However, the observation of only two Tb modes suggests a simple two-state orbital model consisting of a uniaxial ground state and an isotropic excited state. Thermal occupancy of the isotropic state weakens the magnetic anisotropy and drives the SR transition. Classical Monte Carlo simulations show that the simultaneous observation of both modes is consistent with a random, quasi-static distribution of these orbital states, suggesting that Tb orbital quantum states have a longer lifetime than the ps time scale of the Tb spin waves.

## Results
### Experimental data
Tb166 has been shown to host a complex spin wave dispersion characterized by large intralayer ferromagnetic Mn-Mn exchange coupling and competing Mn-Mn and Mn-Tb interlayer couplings with excitations that extend beyond 200 meV[24]. Figure 1 shows INS measurements of the low-energy dynamical susceptibility of intralayer excitations along the (H,0,1), (H,0,2), and (H,0,3) directions as a function of energy for different temperatures. At low temperatures (≤100 K), well within the uniaxial ferrimagnetic ground state, two branches are observed at low energies (<30 meV), as shown in Fig. 1a and b. The very steep branch with a spin gap of $\Delta$ = 6.5 meV corresponds to acoustic magnons that propagate principally within the Mn kagome layer. The narrow branch near 25 meV corresponds to spin waves propagating within the Tb triangular layer (upper Tb mode).

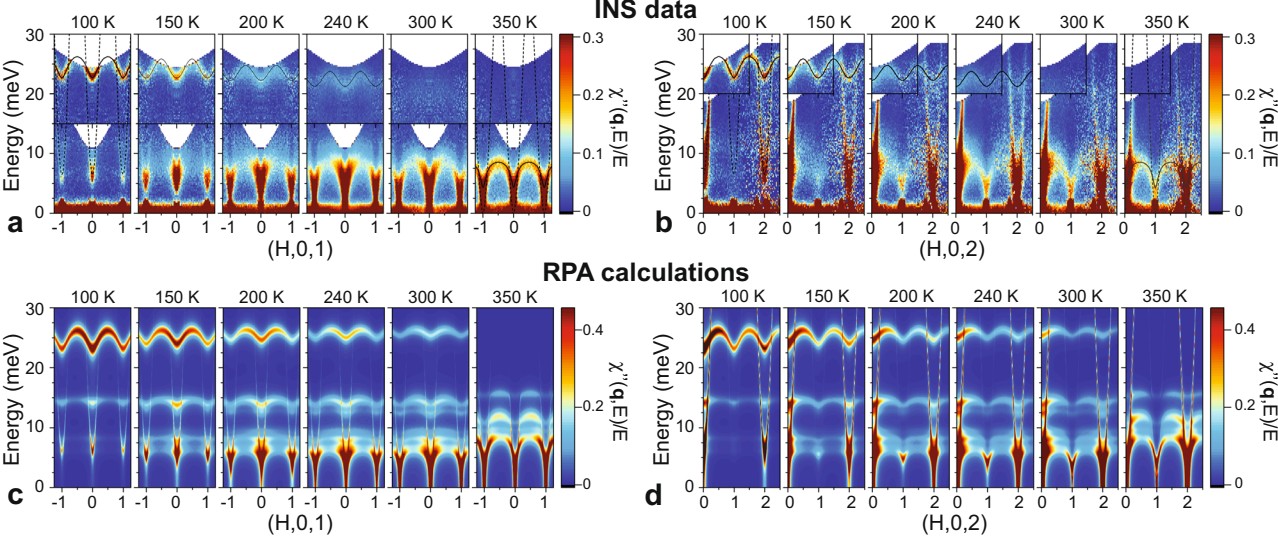

**Fig. 1 | Temperature dependence of the low-energy spin excitations in TbMn₆Sn₆.** The evolution of the low-energy ferrimagnetic spin waves, plotted as $\chi''(\mathbf{q}, E)/E$. Inelastic neutron scattering (INS) data are shown along the **a** (H, 0, 1) and **b** (H, 0, 2) directions at T = 100, 150, 200, 240, 300, and 350 K through the spin-reorientation (SR) transition at $T_{SR}$ = 310 K. Insets at higher energy in **a** and **b** correspond to data along (H, 0, 3). At 100 K, the dashed and solid lines correspond to the ground state dispersion as obtained from fits to the Heisenberg model described in Ref. 24. Calculations of the spin dynamics in the random-phase approximation (RPA) are shown along the **c** (H, 0, 1) and **d** (H, 0, 2) directions at the same temperatures as the data. In **a** and **b** in the SR-phase at 350 K, the dashed and solid lines correspond to the RPA dispersions obtained using the Heisenberg parameters and a magnetically isotropic Tb ion. Thinner lines at other temperatures are guides to the eye.

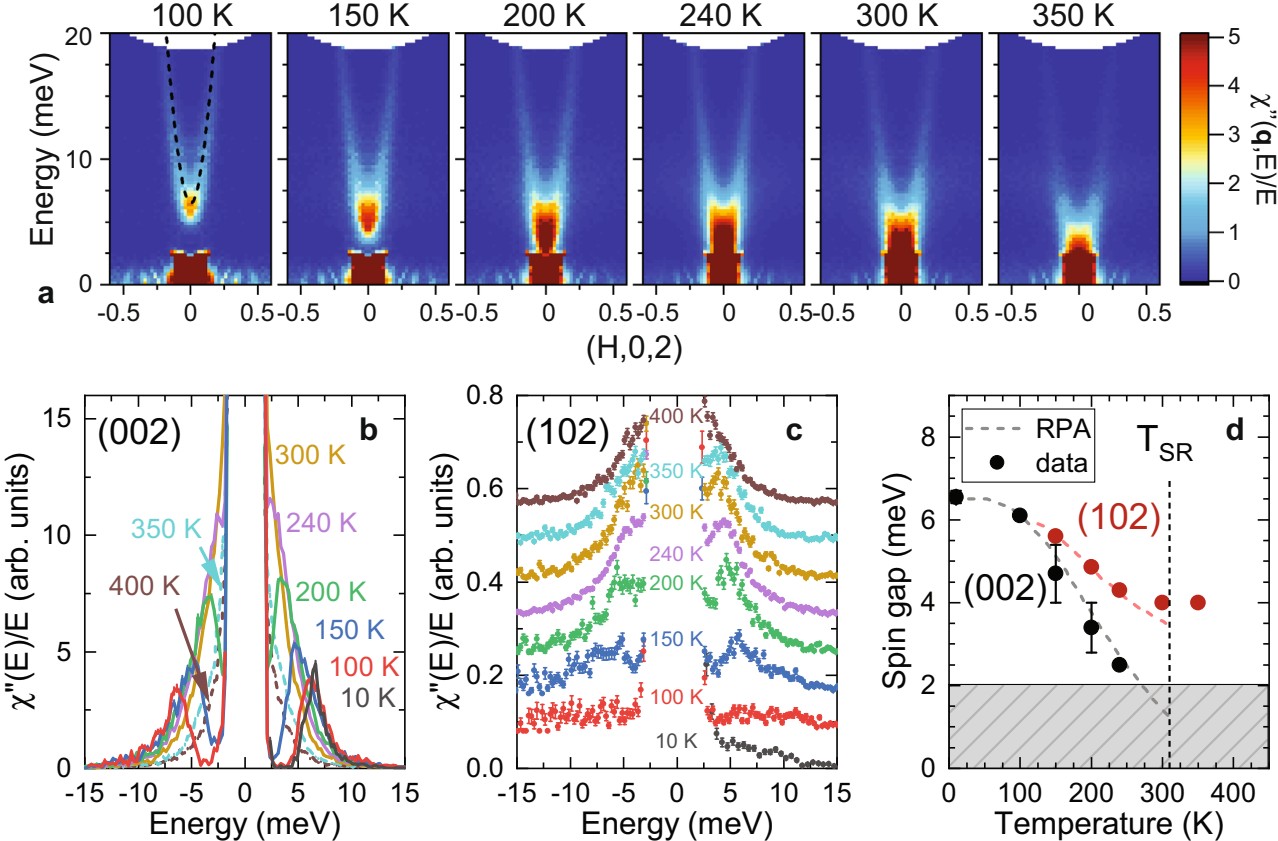

**Fig. 2 | Temperature dependence of the spin gap in TbMn₆Sn₆. a** The evolution of the acoustic mode intralayer spin wave dispersion along $\mathbf{q} = (H, 0, 2)$ at $T = 100$, 150, 200, 240, 300, and 350 K through the spin-reorientation (SR) transition at $T_{SR} = 310$ K. The dashed line at $T = 100$ K corresponds to the ground state dispersion as obtained from fits to the Heisenberg model described in Ref. 24. **b** The inelastic neutron scattering spectra at (0,0,2) are plotted as $\chi''(\mathbf{q}, E)/E$ for several temperatures up to (solid lines) and above $T_{SR}$ (dashed lines). **c** INS spectra at (1,0,2) for several temperatures. **d** The temperature dependence of the spin gap energy at (0, 0, 2) and (1, 0, 2) as obtained from the maxima in panels **b** and **c**. The dashed lines in **d** correspond to random-phase approximation (RPA) calculations in the uniaxial phase. Observation of dynamical features in the hashed area are obscured by the instrumental elastic resolution. Error bars are statistical in origin and represent one standard deviation.

Upon increasing the temperature towards $T_{SR}$, Fig. 1a and b show that several phenomena are observed. First, the spin gap gradually closes to zero at $T_{SR}$. Second, the Tb mode at 25 meV begins to soften and its intensity is strongly reduced, vanishing at $T_{SR}$. Third, broad excitations associated with orbital excitons begin to appear below 10 meV at 150 K and grow in strength. The low-energy features coalesce to form a collective spin wave branch between 5 meV and 10 meV above $T_{SR}$ which we identify as the lower Tb mode.

Figure 2 shows the evolution of the spin gap at $\mathbf{q} = (0,0,2)$, where the intensity of the Mn acoustic magnon is very strong, upon approach to the SR transition. Up to 100 K, the low-energy dispersion is largely the same as the ground state dispersion at $T = 7$ K reported in Ref. 24 and is shown as the dashed line in Fig. 2a. Above 100 K, the spin gap begins to close. Figure 2b shows the low-energy spectra cut through (0,0,2) that illustrate more clearly that the gap follows a roughly linear reduction above 100 K (Fig. 2d). As described below, the thermally-driven reduction of the spin gap is associated with spin fluctuations that decrease the average Tb sublattice magnetization, consistent with neutron diffraction measurements[25], and should become gapless in the easy-plane phase above $T_{SR}$. Figure 2c and d show that the lower Tb mode develops with a significant gap at (1,0,2) of almost 6 meV at 150 K, which softens as $T_{SR}$ is approached but retains a 4 meV gap in the SR-phase at 350 K.

Figure 2b shows that warming above $T_{SR}$ (from 300 K to 350 K) leads to an immediate two-fold reduction of the susceptibility. As the dipole scattering factor for INS measures the

magnetic moment (**m**) components perpendicular to $\mathbf{q} = (0,0,2)$, this reduction reflects the change in the moment direction on the transverse spin fluctuations since $\mathbf{m} \parallel \hat{z}$ gives $\chi''(T < T_{SR}) = \chi''_{xx} + \chi''_{yy} = 2\chi''_{xx}$ and $\mathbf{m} \perp \hat{z}$ gives $\chi''(T > T_{SR}) = \chi''_{xx}$.

Figure 3a shows the temperature dependence of the Tb modes through the *M*-point of the hexagonal Brillouin zone boundary at (1/2, 0, L) as a function of L. At the zone boundary, the acoustic Mn mode is absent and the Tb modes along this direction are dispersionless, allowing an unobstructed observation of the crossover from the Tb upper mode to the lower mode with the coexistence of the two modes at intermediate temperatures. This is more clearly seen in Fig. 3b, which displays the dynamical susceptibility at (1/2,0,3) as a function of temperature and energy.

The susceptibility of the upper and lower Tb modes has a minimum at $(1/2, 0, L = 0)$ when $T < T_{SR}$, as shown in Fig. 3a and d. This minimum is again a consequence of the sensitivity of the dipole factor to the Tb moment direction ($\mathbf{m}_{Tb}$). Calculations of the dipole factor times the Tb magnetic form factor clearly indicate that both the upper and lower Tb modes are transverse excitations with $\mathbf{m}_{Tb} \parallel \hat{z}$ when $T < T_{SR}$. Upon warming above $T_{SR}$, Fig. 3a and e indicates a susceptibility maximum at (1/2, 0, 0), consistent with the form factor for transverse excitations of an in-plane moment configuration with $\mathbf{m}_{Tb} \perp \hat{z}$.

## Model Hamiltonian

The evolution of the upper and lower Tb modes is similar to the appearance of orbital excitons that are reported to occur in other

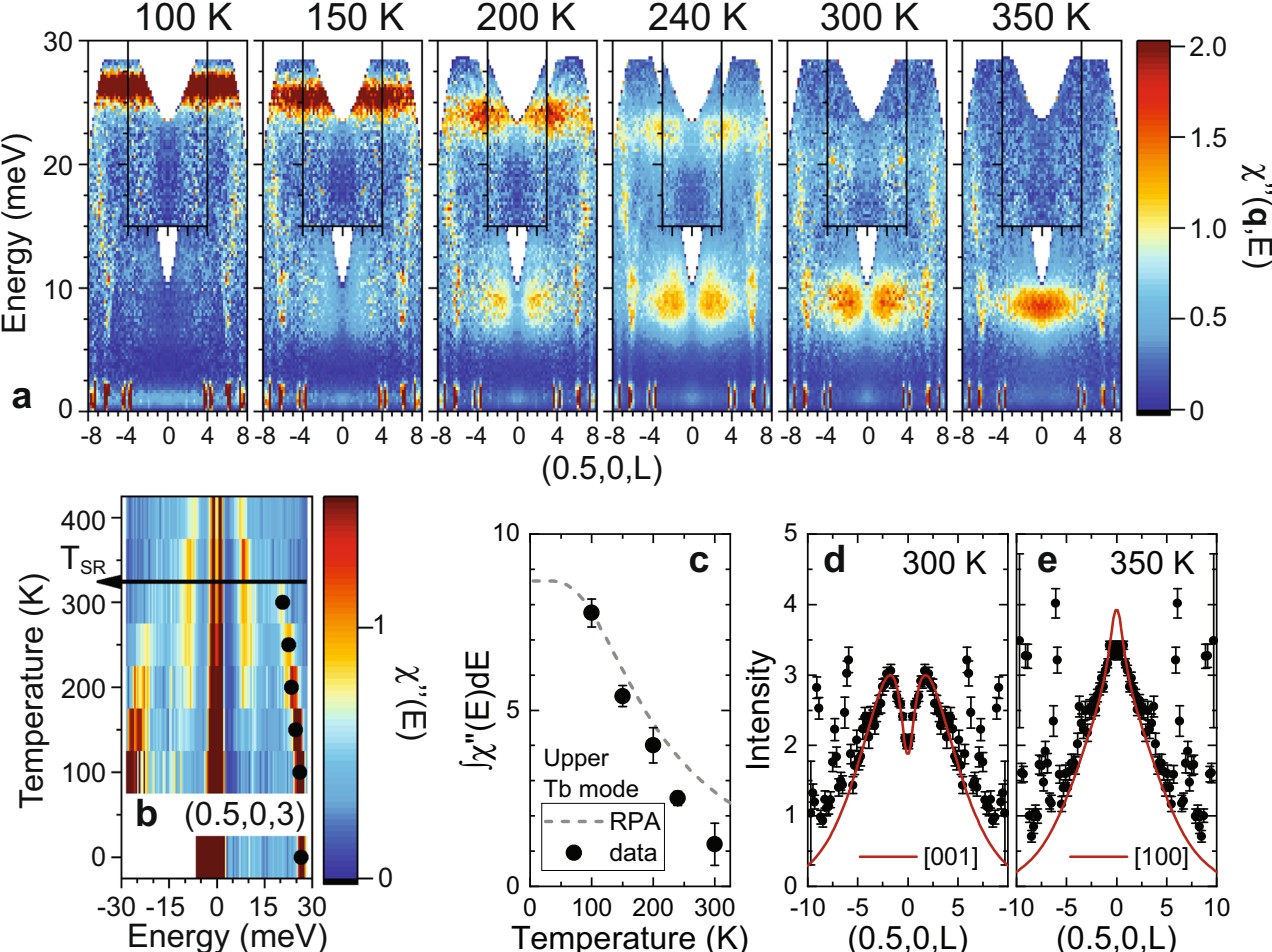

**Fig. 3 | Temperature dependence of the zone-boundary spin excitations in TbMn$_6$Sn$_6$. a** The evolution of the upper and lower Tb spin wave modes along the $(1/2, 0, L)$ direction at $T = 100, 150, 200, 240, 300,$ and $350$ K through the spin-reorientation (SR) transition at $T_{SR} = 310$ K. Inset shows data along the $(3/2, 0, L)$ direction. **b** The inelastic neutron scattering data at $(1/2,0,3)$ as a function of energy and temperature with intensities plotted as $\chi''(E)$. The filled symbols indicate that the energy of the upper mode drops in a similar fashion to the spin gap. **c** The

integrated susceptibility of the upper Tb mode as a function of temperature compared to random-phase approximation (RPA) calculations. **d, e** The structure factor of the lower mode at 6 meV along $(1/2, 0, L)$ at 300 K and 350 K, respectively. The solid red lines are calculations of the dipole factor times the Tb magnetic form factor for transverse excitations with $\mathbf{m}_{Tb}\|[0,0,1]$ (**d**) and $\mathbf{m}_{Tb}\perp[0, 0, 1]$ (**e**) averaged over equivalent domains. Sharp excitations seen at $L = \pm 6$ in panels **a, d**, and **e** are phonons. Error bars are statistical in origin and represent one standard deviation.

rare-earth magnets, such as PrTl$_3$[26,27] and TbSb[28], and also transition metal compounds[29,30]. These excitons arise from the thermal population of low-lying CEF states which are predominantly local in character, but propagate from site-to-site due to exchange coupling. Here, we analyze the spin excitations in Tb166 using a similar RPA approach as outlined in the above references.

A minimal description of the magnetic Hamiltonian is given by $\mathcal{H} = \mathcal{H}_{Tb} + \mathcal{H}_{Mn} + \mathcal{H}_{ex}$ and includes terms that describe the local CEF states of the Tb ion with total angular momentum $J = 6$ and spin $S = 3$ ($\mathcal{H}_{Tb}$), the single-ion anisotropy of the Mn ion with spin $s = 1$ ($\mathcal{H}_{Mn}$), and the isotropic (Heisenberg) exchange couplings within and between the Mn-Mn and Mn-Tb magnetic sublattices ($\mathcal{H}_{ex}$).

The Heisenberg Hamiltonian is

$$\mathcal{H}_{ex} = \sum_{i,j} \mathcal{J}_{ij}^{MM} \mathbf{s}_i \cdot \mathbf{s}_j + \mathcal{J}^{MT} \sum_{\langle i<j \rangle} \mathbf{s}_i \cdot \mathbf{S}_j \qquad (1)$$

where $\mathcal{J}_{ij}^{MM}$ represent various, and quite strong, intralayer and interlayer magnetic couplings between Mn spins ($\mathbf{s}$), as described in detail in Ref. 24. Here, $\mathcal{J}^{MT} > 0$ is the AF coupling between neighboring Mn and Tb spins ($\mathbf{S}$) that result in tightly bound Mn-Tb-Mn collinear ferrimagnetic trilayers.

DFT calculations[16] and the analysis of magnetization data near the SR-transition[19,21] suggest that a Tb CEF Hamiltonian

$$\mathcal{H}_{Tb} = B_2^0 \mathcal{O}_2^0 + B_4^0 \mathcal{O}_4^0 \qquad (2)$$

where $\mathcal{O}_l^0$ are the Steven's operators, is sufficient to explain the SR transition. Figure 4a shows the level scheme for arbitrary CEF parameters using Segal and Wallace notation[31] where $B_2^0 = W(1 - |y|)$ and $60B_4^0 = Wy$. The CEF states are labeled by $|m_J\rangle$ which are good quantum numbers for $\mathcal{H}_{Tb}$. Determination of the CEF and exchange parameters and a figure showing the magnetic structure and interactions can be found in Supplementary Notes 1 and 2 and Supplementary Fig. 1.

In Fig. 4a, the $|\pm 6\rangle$ CEF ground state is consistent with the uniaxial character of the Tb ion at low temperatures. We now consider the local Tb orbital dynamics by treating $\mathcal{H}_{ex}$ within a mean-field approximation. A molecular field Hamiltonian acting on the Tb site is given by $\mathcal{H}_{MF}^{Tb} = B_{MF}^{Tb} S_z$, where $B_{MF}^{Tb} = 12\mathcal{J}^{MT} \langle s_z \rangle$ and points along the $-c$-axis (opposite to the Tb moment direction) in the ferrimagnetic ground state. $\mathcal{H}_{MF}^{Tb}$ creates a Zeeman splitting of $|\pm m_J\rangle$ CEF states, as shown in Fig. 4b. The Heisenberg parameters in Supplementary Table 1 provide $B_{MF}^{Tb} = 22.2$ meV at low temperatures, as indicated by the vertical

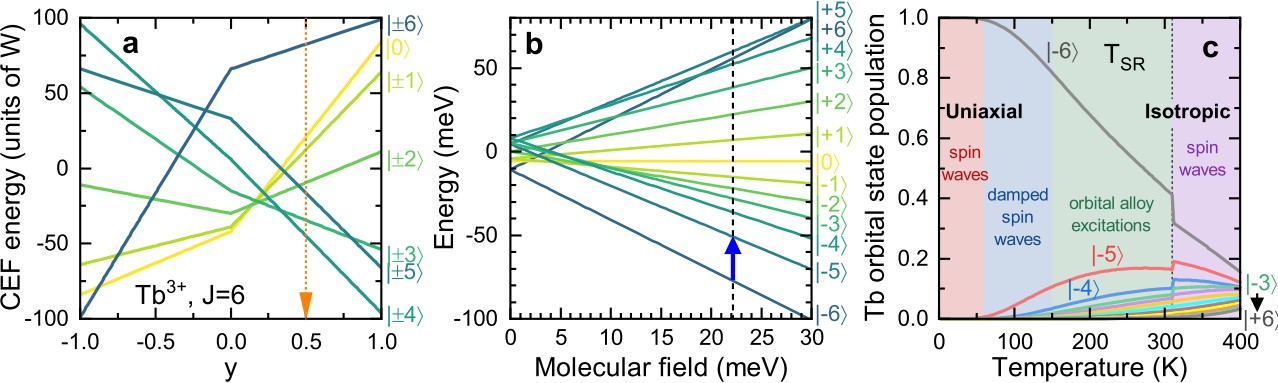

**Fig. 4 | Crystal field model for TbMn₆Sn₆.** (**a**) Crystal electric field (CEF) energy levels for a $J = 6$ Tb ion with dominant $B_2^0$ and $B_4^0$ terms. The orange arrow indicates the experimental values with $B_2^0 = -0.0347$ meV and $B_4^0 = -0.00143$ meV. **b** The Zeeman splitting of the CEF levels due to the molecular field acting on the Tb ion. The vertical line shows the estimated ground state molecular field strength of $B_{MF}^{Tb} = 22.2$ meV and the blue arrow labels the only dipole-allowed transition out of the ground state. Orbital states are labeled as $|m_j\rangle$. **c** Tb state populations as a function of temperature, where the molecular field decreases according to the measured Mn sublattice magnetization[25]. $T_{SR}$ indicates the spin-reorientation transition.

dashed line in Fig. 4b. The large molecular field results in a Zeeman-like spectrum that is distorted by the CEF potential.

In typical magnets, increasing the temperature leads to fluctuations that decrease the average magnetization, causing a concomitant decrease in the molecular field. What is unique about Tb166 is that the Mn sublattice magnetization initially drops very slowly with increasing temperature[25] due to the strong $\mathcal{J}^{MM}$ couplings. As a result, the strength of the molecular field is maintained while orbital excited states on the Tb ion become thermally populated. As shown in Fig. 4b and c, this initially leads to thermal occupancy of the $|-5\rangle$ state, which deviates from the linear spin-wave theory approximation and should result in damping, as observed in the upper Tb mode beginning around 100 K. As the temperature is increased further, thermal depopulation of the $|-6\rangle$ ground state becomes severe, reaching only about 50% occupancy at 250 K, which explains the weakening of the upper Tb mode intensity that is comprised of excitations out of the ground state. The thermally excited CEF states will also generate orbital excitons which disperse due to $\mathcal{J}^{MT}$.

### RPA calculations
We can test these ideas using a mean-field RPA approach,[27,28] which couples thermally excited CEF levels through the magnetic exchange interactions listed in Supplementary Table 1. Figure 1c and d shows results from RPA calculations of the dynamical susceptibility at various temperatures along the $(H, 0, 1)$ and $(H, 0, 2)$ directions. This also includes RPA calculations performed in the easy-plane state at $T > T_{SR}$ where the molecular field lies in the $ab$-plane, resulting in a different spectrum of local Tb states.

The RPA results demonstrate several of the observed features described above, including closure of the spin gap [see Figs. 1 and 2d], decreasing strength of the upper Tb mode (Fig. 3c), and the appearance of orbital excitons below 10 meV. These comparisons also reveal several limitations of the RPA approach. Most notably, RPA predicts narrowing (reduced dispersion) and hardening of the upper Tb mode with increasing temperature, whereas the INS data indicate narrowing and softening along with the typical development of lifetime broadening. One source of discrepancy is that RPA underestimates the decrease of the Mn sublattice magnetization with temperature, thereby overestimating the molecular field and its temperature dependence. More importantly, RPA predicts the appearance of many modes (eg. near 15 meV), whereas only two Tb modes are observed in the INS data. These disagreements are also likely a limitation of RPA, which truncates higher-order mode-mode coupling terms that result in decay and which should become

increasingly important at higher temperatures when Tb thermal fluctuations are large.

### Orbital quantum alloy model
Here, we define a simple effective two-state orbital model that ensures the appearance of only two Tb modes. The ground state singlet $\Psi_u$ has uniaxial anisotropy and the excited state $\Psi_i$ is highly degenerate ($2J = 12$ states) and magnetically isotropic, as shown in Fig. 5b. Normally, magnetic isotropy is recovered when all CEF states have equal occupancy, most commonly occurring from Boltzmann statistics at high temperatures (see Fig. 4c). Whereas a uniaxial ground state $\Psi_u$ describes the low-temperature data consisting only of the upper Tb mode, the assumption of completely isotropic Tb ions describes the SR-phase quite well, where only the lower Tb mode is found in RPA calculations, as shown in Fig. 5a. After the renormalization of $\mathcal{J}^{MT}$ because of reduction of the Mn sublattice magnetization by 20%, excellent agreement with the INS data in the SR-phase is achieved. In this two-state model, the energy difference of the upper and lower mode is just the anisotropy energy, $E_{\Psi_u} - E_{\Psi_i} \approx 2K_1/J = 15$ meV, where $K_1$ is the Tb anisotropy constant determined from INS data (see Supplementary Note 2).

To explain the INS data at intermediate temperatures where both modes are observed, the orbitally isotropic excited state must be present below $T_{SR}$. At a given instant, a Tb ion can be found in either of two different orbital quantum states, $\Psi_u$ and $\Psi_i$, respectively, as shown in Fig. 5b, with a "concentration" determined by Boltzmann statistics. These orbital states must be long-lived on the ~THz time-scale of the spin wave frequencies, essentially behaving as a quenched magnetic binary alloy with mixed anisotropy (e.g., such as a Tb and Gd substitutional alloy). Surprisingly, classical Monte Carlo spin dynamics simulations of a quenched magnetic alloy capture the essential features of our observations as the concentration of the alloy changes, as shown in Fig. 5c.

### Discussion
Microscopic CEF and magnetic exchange parameters obtained from INS data confirm an intuitive picture of the SR-transition in Tb166 where thermal softening of the uniaxial Tb anisotropy eventually drops below the Mn easy-plane anisotropy energy. INS data also reveal details of how the orbital dynamics drive this transition. RPA calculations that account for local Tb CEF orbital states and their exchange coupling can capture many essential features of the observed orbital dynamics that result in the closing of the spin gap, the disappearance of the upper Tb mode, and the appearance of dispersive low-energy

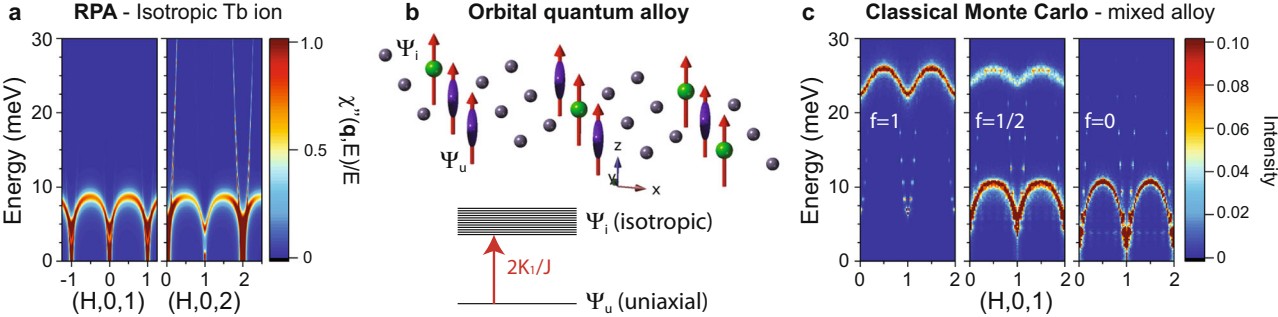

**Fig. 5 | Quantum alloy model for TbMn$_6$Sn$_6$. a** Random-phase approximation (RPA) calculations of the excitations in the spin-reorientation phase for an isotropic Tb ion with $B_2^0 = B_4^0 = 0$. **b** Instantaneous configuration of a quantum alloy on the Tb triangular lattice. Magnetically uniaxial ($\Psi_u$) and isotropic ($\Psi_i$) quantum states of the Tb ion are split by the magnetic anisotropy energy ($2K_1/J$). **c** Classical Monte Carlo simulations of a quenched mixed anisotropy alloy with uniaxial fractions $f = 1$, 1/2, and 0.

excitations comprised of orbital excitons. The RPA calculations outline a rich spectra of these excitons which belie a much simpler observed spectrum only consisting of coexisting upper and lower Tb modes. We can reconcile this simpler observation by implementing a two-state orbital model for Tb, where a temperature-dependent fraction, $f(T)$, of the Tb ions are magnetically uniaxial and $1 - f(T)$ are isotropic on average. The isotropic state can be considered as a quasi-classical coherent state formed from a superposition of angular momentum states (ostensibly due to mode-mode coupling) which replaces the discrete CEF level spectrum of excited states with a single, highly degenerate level. In this scenario, the material behaves as a two-state quantum orbital alloy.

For thermodynamic properties, this distinction has no consequences due to ensemble averaging. We predict, for example, that the SR transition occurs at $f(T_{SR}) \approx 30\%$ when the average magnetic anisotropy of Tb and Mn ions is zero (see Fig. 4c and Supplementary Notes 2 and 3 and Supplementary Figs. 4 and 5). This is in rough correspondence with the appearance of the SR-phase in Gd$_{1-x}$Tb$_x$Mn$_6$Sn$_6$ mixed anisotropy alloys with $x \approx 0.2$[32].

However, there are essential differences for the dynamical properties (such as spin waves) which are fast on the time scale of the orbital fluctuations. On these time scales, the quantum alloy behaves as a quenched random alloy and translational symmetry is broken by the instantaneous configuration of orbital states. In topological electronic materials, electronic time scales are very fast, and band structure calculations should consider quasi-static quantum alloy configurations which include broken translational symmetry. Whereas a classical binary alloy, such as Gd$_{1-x}$Tb$_x$Mn$_6$Sn$_6$, is only found in one of these random configurations and its phase diagram will be affected by percolation and domain effects, the quantum alloy will explore all configurations over time. All quantum materials properties must account for the configuration of quantum states in some way (e.g. Born-Oppenheimer and frozen phonon approximations), but Tb166 is special, since there are effectively two distinct magnetic/orbital quantum states. This makes Tb166 a simple, magnetic quantum binary alloy. More generally, the tunable state configuration of quantum alloys (with temperature, for example) can be used to accelerate our understanding of how disorder affects magnetic and electronic states under adiabatic conditions.

## Methods
### Neutron scattering
Tb166 crystallizes in the HfFe$_6$Ge$_6$-type structure with hexagonal space group P6/mmm (No. 191) and Mn, Sn1, Sn2, Sn3, and Tb ions, respectively, sitting at the 6i, 2e, 2d, 2c, and 1b Wyckoff positions, as shown in Supplementary Fig. 1. Neutron diffraction data were taken on the BT7 triple-axis spectrometer at the NIST Center for Neutron

Research with a PG(002) monochromator and analyzer fixed at an energy of 14.7 meV ($\lambda = 2.359$ Å)[33]. Vertical focusing was employed for the monochromator and both the monochromator and analyzer were in the flat horizontal focusing configuration. PG filters were used before and after the sample to reduce $\lambda/2$ neutrons in the beam. The Söller collimation before the monochromator, before the sample, after the sample, and before the analyzer were 25'-50'-50'-25' full-width-at-half maximum, respectively. The sample used was a 30 mg single crystal with an 8' intrinsic mosaic. Momentum transfers are reported in hexagonal coordinates as $\mathbf{q}[r.l.u.] = (H, K, L)$ where $q[\text{Å}^{-1}] = (4\pi H/(a\sqrt{3}), 4\pi K/(a\sqrt{3}), 2\pi L/c)$. The sample was sealed in an aluminum can with helium exchange gas to promote temperature uniformity and was aligned in the reciprocal space $(H, H, L)$ scattering plane in hexagonal notation.

The onset of magnetic order with propagation vector $\mathbf{k} = (0, 0, 0)$ was observed at $T_C = 420$ K by the increase in intensity at integer $(H, K, L)$ Bragg peak positions, as shown in Supplementary Fig. 2 and in agreement with previous neutron diffraction measurements[7]. The spin-reorientation transition, $T_{SR}$, can be seen just above 300 K as the sudden decrease in intensity of this and other $(0, 0, L)$-type Bragg peaks. Hysteresis was observed about $T_{SR}$, and to ensure this was not an artifact from the sample not equilibrating, a slow temperature sweep of 0.1 K/min was used both on cooling and warming at the (0,0,1) Bragg peak. As the sample is cooled through $T_{SR}$, the moments, which are within the ab-plane above $T_{SR}$, rotate towards the c-axis below $T_{SR}$ adopting the magnetic space group P6/mm'm' (No. 191.240) where both the Tb and Mn magnetic sublattices again are antiparallel but are confined to the c-axis (see Supplementary Fig. 1). This results in a sudden decrease of the (0,0,1) Bragg peak because neutrons are only sensitive to the components of ordered magnetic moments which are perpendicular to the scattering vector, $\mathbf{q}$. The SR-transition extinguishes the intensity as the moments orient parallel to $\mathbf{q} = (0, 0, L)$, as shown in Supplementary Fig. 2(c) and (d). Additional scans along high symmetry directions were performed at 400 K, 300 K, and 150 K to check for the appearance of unexpected commensurate or incommensurate magnetic Bragg peaks away from integer $(H, K, L)$ positions and no new peaks were observed. Below $T_{SR}$ the temperature evolution of the Bragg peaks, such as (1,1,1) peak shown in Supplementary Fig. 2(a), evolve smoothly within the uniaxial ferrimagnetic ground state.

Inelastic neutron scattering was performed on the Wide Angular-Range Chopper Spectrometer (ARCS) located at the Spallation Neutron Source at Oak Ridge National Laboratory using single crystals of Tb166 grown from excess Sn using the flux method, with details described in Ref. 24. An array of 9 crystals with a total mass of 2.564 grams was co-aligned with the hexagonal $(H, 0, L)$ scattering plane set horizontally, and attached to a top-loading closed-cycle refrigerator.

The data were collected at various temperatures using an incident energy of $E_i = 30$ meV with an elastic resolution full-width-at-half-maximum of 1.4 meV. The sample was rotated around the vertical axis to increase the **q** coverage.

The INS data are presented in terms of the orthogonal hexagonal vectors (1,0,0) and (−1,2,0). The measured INS intensities are proportional to the spin-spin correlation function $S(\mathbf{q},E) = \sum_{\alpha\beta}(\delta_{\alpha\beta} - \hat{q}_\alpha\hat{q}_\beta) S_{\alpha\beta}(\mathbf{q},E)$, where $E$ is the energy transfer and $\hat{q}_\alpha$ is the cartesian component of the (unit) momentum transfer vector. In order to remove the trivial temperature dependence caused by the thermal population factor, the INS data are reported as the dynamical susceptibility $\chi''(\mathbf{q},E) = S(\mathbf{q},E)[1 - \exp(-E/k_BT)]^{-1}$ or as $\chi''(\mathbf{q},E)/E$. To improve statistics, the data have been symmetrized with respect to the crystallographic space group $P6/mmm$.

### Random-phase approximation calculations

The RPA approach begins by considering the local-ion Green's functions

$$g_{\alpha\beta}^i(\omega) = \sum_{m,n} \frac{(f_m - f_n)\langle n|J_\alpha^i|m\rangle\langle m|J_\beta^i|n\rangle}{\omega - i\gamma + \epsilon_m - \epsilon_n}. \quad (3)$$

where $|n\rangle, \epsilon_n, f_n$ are the eigenstates, eigenvalues, and thermal occupancies obtained from the local-ion Hamiltonians $\mathcal{H}_i + \mathcal{H}_{MF}^i$ for $i$ = Tb or Mn, respectively. The sites are coupled through the pairwise exchange interactions, giving a set of coupled linear equations for the propagating Green's function

$$G_{\alpha\beta}^{ij}(\mathbf{q},\omega) = g_{\alpha\beta}^i(\omega)\delta_{ij} + \sum_k \mathcal{J}^{ik}(\mathbf{q})\sum_\gamma g_{\alpha\gamma}^i(\omega)G_{\gamma\beta}^{kj}(\mathbf{q},\omega) \quad (4)$$

where $\mathcal{J}^{ik}(\mathbf{q}) = \sum_{lmn}\mathcal{J}^{ik}\exp[-i\mathbf{q}\cdot(\mathbf{d}_i - \mathbf{d}_k - \mathbf{R}_{lmn})]$. The dynamical magnetic susceptibility (weighted by the magnetic cross-section) is related to the summation of the imaginary part of the pairwise Green's functions,

$$\chi''_{\alpha\beta}(\mathbf{q},\omega) \propto (\delta_{\alpha\beta} - \hat{q}_\alpha\hat{q}_\beta) \\ \times \sum_{ij}[g^iF^i(\mathbf{q})][g^jF^j(\mathbf{q})]\mathcal{I}m[G_{\alpha\beta}^{ij}(\mathbf{q},\omega)]. \quad (5)$$

Here $g^i$ and $F^i(\mathbf{q})$ are the Lande $g$-factor and magnetic form factor, respectively. The unpolarized INS data measures the summation of $\chi''_{\alpha\beta}$ over $\alpha$ and $\beta$.

### Classical Monte Carlo simulations

Classical Monte Carlo (MC) simulations were performed using the `UppASD` package[34]. $31 \times 31 \times 4$ unit cells containing one Tb and six Mn atoms form the magnetic lattice, giving a total of 26908 spins. We treat the Tb sites as a random alloy with a concentration of $f$ sites having a uniaxial single-ion anisotropy of $D^T = -1.28$ meV and $1 - f$ sites having an anisotropy of $D^T = 0$. All Mn sites have an easy-plane anisotropy of $D^M = 0.44$ meV. A uniaxial collinear ferrimagnetic phase was prepared and annealed at 10 K in 40,000 Monte Carlo steps. Spin dynamics simulations were performed for different alloy concentrations, averaging over 10 different ensembles, using Landau-Lifshitz-Gilbert dynamics with 40,000 Monte Carlo time steps of $3 \times 10^{-16}$ seconds and correlation functions $S(\mathbf{r}, t)$ evaluated over 2000 time steps of $6 \times 10^{-15}$ seconds. The correlation functions are Fourier transformed to obtain $S(\mathbf{q}, \omega)$. Due to the essential difference between classical and quantum treatment of local Tb states and the resultant spin fluctuations, all MC simulations are done at 10 K. All other simulation parameters are shown in Supplementary Table 1.

## Data availability

Source data for line and scatter plots are provided in this paper. Inelastic neutron scattering data analyzed here can be obtained in the MDF open data repository[35,36] with the identifiers https://doi.org/10.18126/VWAE-PA0M and URL 10.18126/VWAE-PA0M. Associated analysis and reduction scripts are available from R.J.M. upon reasonable request. Neutron diffraction are available from R.J.M. upon reasonable request.

## Code availability

Codes for calculating the spin excitations in $TbMn_6Sn_6$ in the random-phase approximation are available from R.J.M. upon reasonable request.

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

## Acknowledgements

The authors would like to acknowledge useful discussions and support from Liqin Ke, Igor Mazin, Allen Scheie, and Stephen Wilson. R.J.M., B.G.U., T.H., B.L., and S.X.M.R.'s work at the Ames Laboratory is supported by the U.S. Department of Energy (USDOE), Office of Basic Energy Sciences, Division of Materials Sciences and Engineering. T.J.S. and P.C.C. are supported by the Center for the Advancement of Topological Semimetals (CATS), an Energy Frontier Research Center funded by the USDOE Office of Science, Office of Basic Energy Sciences, through the Ames National Laboratory. Ames National Laboratory is operated for the USDOE by Iowa State University under Contract No. DE-AC02-07CH11358. A portion of this research used resources at the Spallation Neutron Source, which is a USDOE Office of Science User Facility operated by the Oak Ridge National Laboratory. Crystal growth and properties characterization work at George Mason University was supported by the U.S. Department of Energy, Office of Science, Basic Energy Sciences, Materials Science, and Engineering Division.

## Author contributions

R.J M., B.G.U., S.X.M.R., B.L., T.H., and D.L.A. conducted and analyzed INS experiments. J.W.L. and R.L.D. conducted and analyzed neutron diffraction measurements. R.J.M., P.M.S., C.S., and H.L. developed code for RPA calculations. R.J.M. performed classical Monte Carlo simulations. P.C.C. and T.J.S. grew and characterized single crystals for INS measurements. N.G. and H.B. grew and characterized single crystals for neutron diffraction measurements.

## Competing interests

The authors declare no competing interests.
