## [Peer Review File · Nature Communications]

Reviewers' Comments:

Reviewer #1:

Remarks to the Author:

Dear Editor,

I have reviewed the manuscript "Orbital character of the spin-reorientation transition in TbMn₆Sn₆" by Riberolles et al. In my opinion the manuscript is technically flawless. Authors have performed detailed analysis of underlying mechanism behind spin-reorientation transition to elucidate the orbital contribution. Also, the topic is timely, which may warrant publication in Nature Comm. That being said, I have two concerns that authors can address:

first, the manuscript is solely written from a condensed matter physics perspective (in fact from neutron scattering perspective). Given the broad dissemination of Nat. Comm., it is strongly recommended that the introduction section of the manuscript be modified to be accessible by non-neutron scatterers as well. In the current form, the scope is quite narrow.

second, authors have emphasized on the topological insulating character of the material quite a few times in the manuscript. But no detail is provided e.g. Chern number, Berry curvature or even some basic information on how orbital order affects the (so called) topological behavior. This issue needs to be addressed.

Overall, I recommend publication of the manuscript.

Reviewer #2:

Remarks to the Author:

TbMn₆Sn₆ is a topological kagome magnet where the out-of-plane ferromagnetic Mn kagome layer hosts emergent two-dimensional Chern-gapped massive Dirac fermion. In this study, the authors report inelastic neutron scattering of the magnetic anisotropy, and the spin reorientation (SR) transition at 310~K in TbMn₆Sn₆. They found the Tb ion has an effective two-state orbital character, with a uniaxial ground state and an isotropic excited state due to a combination of the crystalline electric field and Tb-Mn magnetic coupling. The two states compete with each other and the thermally-driven critical concentration of isotropic Tb ions eventually triggers the SR transition. A spatially-random orbital alloy scenario was discussed to interpret the observation.

As the topological states in TbMn₆Sn₆ are, in principle, sensitive to detailed magnetism, understanding the microscopic origins of the spin excitation is crucial for interpreting and manipulating the quantum magnet. I will recommend the publication if the following questions are answered satisfactorily:

1, Both the inelastic neutron scattering results and the calculations suggest a gradual, thermal-driven crossover from the uniaxial ground state to the isotropic excited state. The authors should explain why this competition triggers a first-order SR transition instead of a crossover.

2, The authors have put forward a quantum alloy picture where the critical fraction of uniaxial Tb for the SR has been estimated. Can this estimation be applied to similar system, for instance, the complete phase diagram of the Gd_{1-x}Tb_xMn₆Sn₆ in Ref.31? This is important as the control of the SR transition may enable topological switching applications, as the author suggested.

3, There are some typos. For instance, on Page 6, Fig. 4(d) should be Fig. 4(c).

Reviewer #1 (Remarks to the Author):

Dear Editor,

I have reviewed the manuscript "Orbital character of the spin-reorientation transition in TbMn₆Sn₆" by Riberolles et al. In my opinion the manuscript is technically flawless. Authors have performed detailed analysis of underlying mechanism behind spin-reorientation transition to elucidate the orbital contribution. Also, the topic is timely, which may warrant publication in Nature Comm. That being said, I have two concerns that authors can address:

first, the manuscript is solely written from a condensed matter physics perspective (in fact from neutron scattering perspective). Given the broad dissemination of Nat. Comm., it is strongly recommended that the introduction section of the manuscript be modified to be accessible by non-neutron scatterers as well. In the current form, the scope is quite narrow.

We have modified the introduction to be of more general interest to non-neutron scatterers. This new introduction is motivated in part by the Reviewer's second question. Please see the response below.

second, authors have emphasized on the topological insulating character of the material quite a few times in the manuscript. But no detail is provided e.g. Chern number, Berry curvature or even some basic information on how orbital order affects the (so called) topological behavior. This issue needs to be addressed.

In Tb166, the large exchange and crystal-field splittings of the Mn bands are the dominant energy scales that determine which Mn orbital states form Dirac cones that lie close to the Fermi energy. These orbitals are preferably 2D in character (eg. $d(x^2-y^2)$) in order to minimize coupling between kagome layers. The presence of spin-orbit coupling (SOC) within the Mn bands is essential for forming a Chern gap at the Dirac point. In the ferromagnetic (FM) case, the Dirac cones are spin-polarized and the spin orientation controls the size of the Chern gap. For 2D Dirac cones, the gap is maximized when Mn spins point parallel to the Mn orbital angular momentum and therefore perpendicular to the kagome layer. It is this latter condition where the rare-earth ion plays a crucial role in controlling the topology of R166 compounds. The SOC and crystal-field potential of the rare-earth ion will determine its magnetic anisotropy energy and strong R-Mn magnetic exchange coupling imprints this magnetic anisotropy on the Mn spin orientation. Thus, any changes in the orbital occupancies of the rare-earth ion (in this case, by temperature) will couple to the Chern gap. In the extreme case of Tb166, a strongly first-order spin reorientation transition from uniaxial to easy-plane anisotropy should effectively close the Chern gap.

This connects to the main conclusions of this paper:

- (1) Thermally-averaged orbital occupancies of the Tb ion gradually weaken its uniaxial magnetic anisotropy, driving global SR transition.
- (2) At intermediate temperatures Tb orbital/spin fluctuations are significant and map onto a novel two-state orbital model. Orbital/spin fluctuations are slow on electronic time scales of the Mn bands, effectively acting as a random orbital alloy. I suspect that this would have deleterious consequences for quantum and topological transport and coherence, but this is an open question. For example, a recent report connects unusual temperature-driven magnetic fluctuations in Tb166 to the anomalous Hall conductivity (see Ref. 25 in the new version).
- (3) Outside of its effect on topological band properties, the "quantum orbital alloy" is novel state of matter in its own right. A rough analogy can be made to other novel two-state quantum alloys, such as mixed valence compounds.

As with many nominal topological insulators including Tb166, the band filling is not ideal and the Fermi energy is not in the band gap. Thus, Tb166 retains metallic character with strong low-field quantum oscillations and large anomalous Hall effect (AHE). Achieving the quantized AHE of a Chern insulator requires methods to move the Fermi energy into a charge neutral condition (where the filled band would

carry an integer Chern number). Methods to control the Fermi energy and carrier concentration, for example by chemical substitution or gating of thin film samples, are an immense challenge that resides the forefront of topological materials research.

Overall, I recommend publication of the manuscript.

We thank the Reviewer for very constructive comments that have significantly improved the impact and readability of the manuscript.

Reviewer #2 (Remarks to the Author):

TbMn₆Sn₆ is a topological kagome magnet where the out-of-plane ferromagnetic Mn kagome layer hosts emergent two-dimensional Chern-gapped massive Dirac fermion. In this study, the authors report inelastic neutron scattering of the magnetic anisotropy, and the spin reorientation (SR) transition at 310~K in TbMn₆Sn₆. They found the Tb ion has an effective two-state orbital character, with a uniaxial ground state and an isotropic excited state due to a combination of the crystalline electric field and Tb-Mn magnetic coupling. The two states compete with each other and the thermally-driven critical concentration of isotropic Tb ions eventually triggers the SR transition. A spatially-random orbital alloy scenario was discussed to interpret the observation.

As the topological states in TbMn₆Sn₆ are, in principle, sensitive to detailed magnetism, understanding the microscopic origins of the spin excitation is crucial for interpreting and manipulating the quantum magnet. I will recommend the publication if the following questions are answered satisfactorily:

1, Both the inelastic neutron scattering results and the calculations suggest a gradual, thermal-driven crossover from the uniaxial ground state to the isotropic excited state. The authors should explain why this competition triggers a first-order SR transition instead of a crossover.

In Supplementary Note 3, we now include calculations of the free energy of Tb₁₆₆ within a mean-field approximation. Supplementary Fig. 5(a) shows the free energy of the uniaxial and easy-plane configurations as a function of temperature, which gives T_{SR} of ~310 K. Supplementary Fig. 5(b) shows the free energy as a function of the magnetization angle for several temperatures. The free energy shows clear first-order character whereby local minima are maintained at $\theta=0$ and 90 degrees with a discontinuous change in the global minimum at T_{SR}.

2, The authors have put forward a quantum alloy picture where the critical fraction of uniaxial Tb for the SR has been estimated. Can this estimation be applied to similar system, for instance, the complete phase diagram of the Gd_{1-x}Tb_xMn₆Sn₆ in Ref.31? This is important as the control of the SR transition may enable topological switching applications, as the author suggested.

I do not think that the quantum alloy picture would have direct correspondence with a real substitutional alloy and this comparison is only notional. To avoid confusion in the response below, I use the chemical formula from Ref. 31 (Ref. 32 in the new version) which assigns x to the fraction of Gd in Tb_{1-x}Gd_xMn₆Sn₆. The simplest approximation for estimating the phase diagram of a substitutional alloy (mean-field) is to calculate the average magnetic anisotropy. This model of the substitutional is closely related to the quantum alloy picture, which associates the fraction of Tb ions in the uniaxial state at any temperature [$1-x=f(T)$] with the ground state occupancy, as shown in Fig. 4(c). Ensemble averaging of the quantum alloy picture would arrive at the same result as the mean-field substitutional model.

In mean-field model for a substitutional alloy, we expect T_{SR} to fall linearly with x and reach T_{SR}=0 at $x_{SR} = 1-6K_{Mn}/(K_1+K_2) = 0.7$ where the average ground state anisotropy is zero. Inspecting Ref. 32, we see that T_{SR} initially falls linearly, but too slowly to be consistent with the simple scenario above. Also, the data in Ref. 31 show that T_{SR} rapidly goes to zero at Gd compositions between $x=0.8$ and 0.9 . Thus, the average anisotropy of the substitutional alloy also underestimates the critical Gd concentration of the SR-transition at zero-temperature.

It is not too surprising that the experimental phase diagram of the Tb/Gd substitutional alloy does not map directly onto the mean-field anisotropy model. This is due to inhomogeneous configurational effects in the substitutional alloy (eg. defects for small x and percolation and the formation of domains at large x) and how they respond to temperature. In this respect, there is only a notional connection between a real alloy and the quantum alloy picture.

3, There are some typos. For instance, on Page 6, Fig. 4(d) should be Fig. 4(c).

We thank the Reviewer for pointing this out. This and other typos have been fixed in the resubmitted manuscript.

Reviewers' Comments:

Reviewer #1:

Remarks to the Author:

Dear Editor,

I have reviewed the revised manuscript and authors responses to the referees comments. I am satisfied with the authors responses. I recommend the publication of the manuscript in Nature Communications.

Yours sincerely,

Reviewer #2:

Remarks to the Author:

I am satisfied with the free energy calculation. The quantum alloy picture predicts a SR transition at 30%, which is roughly observation at $\text{Gd}_{1-x}\text{Tb}_x\text{Mn}_6\text{Sn}_6$ ($x=0.2$). As the authors point out, the percolations and domain effects may affect the phase diagram in the real alloy.

I am fine for the explanation and recommend publication for this manuscript.